# Reweighting Augmented Samples by Minimizing the Maximal Expected Loss

**Mingyang Yi**[1,2]*, **Lu Hou**[3], **Lifeng Shang**[3], **Xin Jiang**[3], **Qun Liu**[3], **Zhi-Ming Ma**[1,2]
[1]University of Chinese Academy of Sciences
`yimingyang17@mails.ucas.edu.cn`
[2]Academy of Mathematics and Systems Science, Chinese Academy of Sciences
`mazm@amt.ac.cn`
[3]Huawei Noah's Ark Lab
`{houlu3,shang.lifeng,Jiang.Xin,qun.liu}@huawei.com`

## Abstract

Data augmentation is an effective technique to improve the generalization of deep neural networks. However, previous data augmentation methods usually treat the augmented samples equally without considering their individual impacts on the model. To address this, for the augmented samples from the same training example, we propose to assign different weights to them. We construct the maximal expected loss which is the supremum over any reweighted loss on augmented samples. Inspired by adversarial training, we minimize this maximal expected loss (MMEL) and obtain a simple and interpretable closed-form solution: more attention should be paid to augmented samples with large loss values (i.e., harder examples). Minimizing this maximal expected loss enables the model to perform well under any reweighting strategy. The proposed method can generally be applied on top of any data augmentation methods. Experiments are conducted on both natural language understanding tasks with token-level data augmentation, and image classification tasks with commonly-used image augmentation techniques like random crop and horizontal flip. Empirical results show that the proposed method improves the generalization performance of the model.

## 1 Introduction

Deep neural networks have achieved state-of-the-art results in various tasks in natural language processing (NLP) tasks (Sutskever et al., 2014; Vaswani et al., 2017; Devlin et al., 2019) and computer vision (CV) tasks (He et al., 2016; Goodfellow et al., 2016). One approach to improve the generalization performance of deep neural networks is data augmentation (Xie et al., 2019; Jiao et al., 2019; Cheng et al., 2019; 2020). However, there are some problems if we directly incorporate these augmented samples into the training set. Minimizing the average loss on all these samples means treating them equally, without considering their different implicit impacts on the loss.

To address this, we propose to minimize a reweighted loss on these augmented samples to make the model utilize them in a cleverer way. Example reweighting has previously been explored extensively in curriculum learning (Bengio et al., 2009; Jiang et al., 2014), boosting algorithms (Freund & Schapire, 1999), focal loss (Lin et al., 2017) and importance sampling (Csiba & Richtárik, 2018). However, none of them focus on the reweighting of augmented samples instead of the original training samples. A recent work (Jiang et al., 2020a) also assigns different weights on augmented samples. But weights in their model are predicted by a mentor network while we obtain the weights from the closed-form solution by minimizing the maximal expected loss (MMEL). In addition, they focus on image samples with noisy labels, while our method can generally be applied to also textual data as well as image data. Tran et al. (2017) propose to minimize the loss on the augmented samples under the framework of Expectation-Maximization algorithm. But they mainly focus on the generation of augmented samples.

---

*This work is done when Mingyang Yi is an intern at Huawei Noah's Ark Lab.

Unfortunately, in practise there is no way to directly access the optimal reweighting strategy. Thus, inspired by adversarial training (Madry et al., 2018), we propose to minimize the maximal expected loss (MMEL) on augmented samples from the same training example. Since the maximal expected loss is the supremum over any possible reweighting strategy on augmented samples' losses, minimizing this supremum makes the model perform well under any reweighting strategy. More importantly, we derive a closed-form solution of the weights, where augmented samples with larger training losses have larger weights. Intuitively, MMEL allows the model to keep focusing on augmented samples that are harder to train.

The procedure of our method is summarized as follows. We first generate the augmented samples with commonly-used data augmentation technique, e.g., lexical substitution for textual input (Jiao et al., 2019), random crop and horizontal flip for image data (Krizhevsky et al., 2012). Then we explicitly derive the closed-form solution of the weights on each of the augmented samples. After that, we update the model parameters with respect to the reweighted loss. The proposed method can generally be applied above any data augmentation methods in various domains like natural language processing and computer vision. Empirical results on both natural language understanding tasks and image classification tasks show that the proposed reweighting strategy consistently outperforms the counterpart of without using it, as well as other reweighting strategies like uniform reweighting.

## 2 RELATED WORK

**Data augmentation.** Data augmentation is proven to be an effective technique to improve the generalization ability of various tasks, e.g., natural language processing (Xie et al., 2019; Zhu et al., 2020; Jiao et al., 2019), computer vision (Krizhevsky et al., 2014), and speech recognition (Park et al., 2019). For image data, baseline augmentation methods like random crop, flip, scaling, and color augmentation (Krizhevsky et al., 2012) have been widely used. Other heuristic data augmentation techniques like Cutout (DeVries & Taylor, 2017) which masks image patches and Mixup (Zhang et al., 2018) which combines pairs of examples and their labels, are later proposed. Automatically searching for augmentation policies (Cubuk et al., 2018; Lim et al., 2019) have recently proposed to improve the performance further. For textual data, Zhang et al. (2015); Wei & Zou (2019) and Wang (2015) respectively use lexical substitution based on the embedding space. Jiao et al. (2019); Cheng et al. (2019); Kumar et al. (2020) generate augmented samples with a pre-trained language model. Some other techniques like back translation (Xie et al., 2019), random noise injection (Xie et al., 2017) and data mixup (Guo et al., 2019; Cheng et al., 2020) are also proven to be useful.

**Adversarial training.** Adversarial learning is used to enhance the robustness of model (Madry et al., 2018), which dynamically constructs the augmented adversarial samples by projected gradient descent across training. Although adversarial training hurts the generalization of model on the task of image classification (Raghunathan et al., 2019), it is shown that adversarial training can be used as data augmentation to help generalization in neural machine translation (Cheng et al., 2019; 2020) and natural language understanding (Zhu et al., 2020; Jiang et al., 2020b). Our proposed method differs from adversarial training in that we adversarially decide the weight on each augmented sample, while traditional adversarial training adversarially generates augmented input samples.

In (Behpour et al., 2019), adversarial learning is used as data augmentation in object detection. The adversarial samples (i.e., bounding boxes that are maximally different from the ground truth) are reweighted to form the underlying annotation distribution. However, besides the difference in the model and task, their training objective and the resultant solution are also different from ours.

**Sample reweighting.** Minimizing a reweighted loss on training samples has been widely explored in literature. Curriculum learning (Bengio et al., 2009; Jiang et al., 2014) feeds first easier and then harder data into the model to accelerate training. Zhao & Zhang (2014); Needell et al. (2014); Csiba & Richtárik (2018); Katharopoulos & Fleuret (2018) use importance sampling to reduce the variance of stochastic gradients to achieve faster convergence rate. Boosting algorithms (Freund & Schapire, 1999) choose harder examples to train subsequent classifiers. Similarly, hard example mining (Malisiewicz et al., 2011) downsamples the majority class and exploits the most difficult examples. Focal loss (Lin et al., 2017; Goyal & He, 2018) focuses on harder examples by reshaping the standard cross-entropy loss in object detection. Ren et al. (2018); Jiang et al. (2018); Shu et al. (2019) use meta-learning method to reweight examples to handle the noisy label problem. Unlike all

these existing methods, in this work, we reweight the augmented samples' losses instead of training samples.

## 3 MINIMIZE THE MAXIMAL EXPECTED LOSS

In this section, we derive our reweighting strategy on augmented samples from the perspective of maximal expected loss. We first give a derivation of the closed-form solution of the weights on augmented samples. Then we describe two kinds of loss under this formulation. Finally, we give the implementation details using the natural language understanding task as an example.

### 3.1 WHY MAXIMAL EXPECTED LOSS

Consider a classification task with $N$ training samples. For the $i$-th training sample $\boldsymbol{x}_i$, its label is denoted as $y_{\boldsymbol{x}_i}$. Let $f_\theta(\cdot)$ be the model with parameter $\theta$ which outputs the classification probabilities. $\ell(\cdot, \cdot)$ denotes the loss function, e.g. the cross-entropy loss between outputs $f_\theta(\boldsymbol{x}_i)$ and the ground-truth label $y_{\boldsymbol{x}_i}$. Given an original training sample $\boldsymbol{x}_i$, the set of augmented samples generated by some method is $B(\boldsymbol{x}_i)$. Without loss of generality, we assume $\boldsymbol{x}_i \in B(\boldsymbol{x}_i)$. The conventional training objective is to minimize the loss on every augmented sample $\boldsymbol{z}$ in $B(\boldsymbol{x}_i)$ as

$$\min_\theta \frac{1}{N} \sum_{i=1}^N \left[ \frac{1}{|B(\boldsymbol{x}_i)|} \sum_{(\boldsymbol{z}, y_{\boldsymbol{z}}) \in B(\boldsymbol{x}_i)} \ell(f_\theta(\boldsymbol{z}), y_{\boldsymbol{z}}) \right], \tag{1}$$

where $y_{\boldsymbol{z}}$ is the label of $\boldsymbol{z} \in B(\boldsymbol{x}_i)$, and can be different with $y_{\boldsymbol{x}_i}$. $|B(\boldsymbol{x}_i)|$ is the number of augmented samples in $B(\boldsymbol{x}_i)$, which is assumed to be finite.

In equation (1), for each given $\boldsymbol{x}_i$, the weights on its augmented samples are the same (i.e., $1/|B(\boldsymbol{x}_i)|$). However, different samples have different implicit impacts on the loss, and we can assign different weights on them to facilitate training. Note that computing the weighted sum of losses of each augmented sample in $B(\boldsymbol{x}_i)$ can be viewed as taking expectation of loss on augmented samples $\boldsymbol{z} \in B(\boldsymbol{x}_i)$ under a certain distribution. When the augmented samples generated from the same training sample are drawn from a uniform distribution, the loss in equation (1) can be rewritten as

$$\min_\theta R_\theta(\mathbb{P}_U) = \min_\theta \frac{1}{N} \sum_{i=1}^N \left[ \mathbb{E}_{\boldsymbol{z} \sim \mathbb{P}_U(\cdot|\boldsymbol{x}_i)} \left[ \ell(f_\theta(\boldsymbol{z}), y_{\boldsymbol{z}}) \right] - \lambda_P \mathbf{KL}(\mathbb{P}_U(\cdot \mid \boldsymbol{x}_i) \parallel \mathbb{P}_U(\cdot \mid \boldsymbol{x}_i)) \right], \tag{2}$$

where the Kullback–Leibler (KL) divergence $\mathbf{KL}(\mathbb{P}_U(\cdot \mid \boldsymbol{x}_i) \parallel \mathbb{P}_U(\cdot \mid \boldsymbol{x}_i))$ equals zero. Here $\mathbb{P}_U(\cdot \mid \boldsymbol{x}_i)$ denotes the uniform distribution on $B(\boldsymbol{x}_i)$. When the augmented samples are drawn from a more general distribution $\mathbb{P}_B(\cdot \mid \cdot)$[1] instead of the uniform distribution, we can generalize $\mathbb{P}_U(\cdot \mid \cdot)$ here to some other conditional distribution $\mathbb{P}_B$.

$$\min_\theta R_\theta(\mathbb{P}_B) = \min_\theta \frac{1}{N} \sum_{i=1}^N \left[ \mathbb{E}_{\boldsymbol{z} \sim \mathbb{P}_B(\cdot|\boldsymbol{x}_i)} \left[ \ell(f_\theta(\boldsymbol{z}), y_{\boldsymbol{z}}) \right] - \lambda_P \mathbf{KL}(\mathbb{P}_B(\cdot \mid \boldsymbol{x}_i) \parallel \mathbb{P}_U(\cdot \mid \boldsymbol{x}_i)) \right]. \tag{3}$$

**Remark 1.** *When $\mathbb{P}_B(\cdot \mid \boldsymbol{x}_i)$ reduces to the uniform distribution $\mathbb{P}_U(\cdot \mid \boldsymbol{x}_i)$ for any $\boldsymbol{x}_i$, since $\mathbf{KL}(\mathbb{P}_U(\cdot \mid \boldsymbol{x}_i) \parallel \mathbb{P}_U(\cdot \mid \boldsymbol{x}_i)) = 0$, the objective in equation (3) reduces to the one in equation (1).*

The KL divergence term in equation (3) is used as a regularizer to encourage $\mathbb{P}_B$ close to $\mathbb{P}_U$ (see Remark 2). From equation (3), the conditional distribution $\mathbb{P}_B$ determines the weights of each augmented sample in $B(\boldsymbol{x}_i)$. There may exist an optimal formulation of $\mathbb{P}_B$ in some regime, e.g. corresponding to the optimal generalization ability of model. Unfortunately, we can not explicitly characterize such an unknown optimal $\mathbb{P}_B$. To address this, we borrow the idea from adversarial training (Madry et al., 2018) and minimize the maximal reweighted loss on augmented samples. Then, the model is guaranteed to perform well under any reweighting strategy, including the underlying optimal one. Specifically, let the conditional distribution $\mathbb{P}_B$ be $\mathbb{P}_\theta^* = \arg\sup_{\mathbb{P}_B} R_\theta(\mathbb{P}_B)$. Our objective is to minimize the following reweighted loss

$$\min_\theta R_\theta(\mathbb{P}_\theta^*) = \min_\theta \sup_{\mathbb{P}_B} R_\theta(\mathbb{P}_B). \tag{4}$$

The following Remark 2 discusses about the KL divergence term in equation (3).

---

[1] In the following, we simplify $\mathbb{P}_B(\cdot \mid \cdot)$ as $\mathbb{P}_B$ if there is no obfuscation.

**Remark 2.** *Since we take a supremum over $\mathbb{P}_B$ in equation (4), the regularizer $\textbf{KL}(\mathbb{P}_B \parallel \mathbb{P}_U)$ encourages $\mathbb{P}_B$ to be close to $\mathbb{P}_U$ because it reaches the minimal value zero when $\mathbb{P}_B = \mathbb{P}_U$. Thus the regularizer controls the diversity among the augmented samples by constraining the discrepancy between $\mathbb{P}_B$ and uniform distribution $\mathbb{P}_U$, e.g., a larger $\lambda_P$ promotes a larger diversity among the augmented samples.*

The following Theorem 1 gives the explicit formulation of $R_\theta(\mathbb{P}_\theta^*)$.

**Theorem 1.** *Let $R_\theta(\mathbb{P}_B)$ and $R_\theta(\mathbb{P}_\theta^*)$ be defined in equation (1) and (4), then we have*

$$R_\theta(\mathbb{P}_\theta^*) = \frac{1}{N}\sum_{i=1}^{N}\left[\sum_{\boldsymbol{z}\in B(\boldsymbol{x}_i)}\mathbb{P}_\theta^*(\boldsymbol{z}\mid\boldsymbol{x}_i)\ell(f_\theta(\boldsymbol{z}),y_{\boldsymbol{z}}) - \lambda_P\mathbb{P}_\theta^*(\boldsymbol{z}\mid\boldsymbol{x}_i)\log\left(|B(\boldsymbol{x}_i)|\mathbb{P}_\theta^*(\boldsymbol{z}\mid\boldsymbol{x}_i)\right)\right], \quad (5)$$

*where*

$$\mathbb{P}_\theta^*(\boldsymbol{z}\mid\boldsymbol{x}_i) = \frac{\exp\left(\frac{1}{\lambda_P}\ell(f_\theta(\boldsymbol{z}),y_{\boldsymbol{z}})\right)}{\sum_{\boldsymbol{z}\in B(\boldsymbol{x}_i)}\exp\left(\frac{1}{\lambda_P}\ell(f_\theta(\boldsymbol{z}),y_{\boldsymbol{z}})\right)} = \text{Softmax}_{\boldsymbol{z}}\left(\frac{1}{\lambda_P}\ell(f_\theta(B(\boldsymbol{x}_i)),y_{B(\boldsymbol{x}_i)})\right), \quad (6)$$

*where $\text{Softmax}_{\boldsymbol{z}}(\frac{1}{\lambda_P}\ell(f_\theta(B(\boldsymbol{x}_i)),y_{B(\boldsymbol{x}_i)}))$ represents the output probability of $\boldsymbol{z}$ for vector $(\frac{1}{\lambda_P}\ell(f_\theta(\boldsymbol{z}_1),y_{\boldsymbol{z}_1}),\cdots,\frac{1}{\lambda_P}\ell(f_\theta(\boldsymbol{z}_{|B(\boldsymbol{x}_i)|}),y_{|B(\boldsymbol{x}_i)|}))$.*

**Remark 3.** *If we ignore the KL divergence term in equation (3), due to the equivalence of minimizing cross-entropy loss and MLE loss (Martens, 2019), the proposed MMEL also falls into the generalized Expectation-Maximization (GEM) framework (Dempster et al., 1977). Specifically, given a training example, the augmented samples of it can be viewed as latent variable, and any reweighting on these augmented samples corresponds to a specific conditional distribution of these augmented samples given the training sample. In the expectation step (E-step), we explicitly derive the closed-form solution of the weights on each of these augmented samples according to (6). In the maximization step, since there is no analytical solution for deep neural networks, following (Tran et al., 2017), we update the model parameters with respect to the reweighted loss by one step of gradient descent.*

The proof of this theorem can be found in Appendix A. From Theorem 1, the loss of it decides the weight on each augmented sample $\boldsymbol{z} \in B_{\boldsymbol{x}_i}$, and the weight is normalized by Softmax over all augmented samples in $B_{\boldsymbol{x}_i}$. The reweighting strategy allows more attention paid to augmented samples with higher loss values. The strategy is similar to those in (Lin et al., 2017; Zhao & Zhang, 2014) but they apply it on training samples.

## 3.2 TWO TYPES OF LOSS

For augmented sample $\boldsymbol{z} \in B(\boldsymbol{x}_i)$, instead of computing the discrepancy between the output probability $f_\theta(\boldsymbol{z})$ and the hard label $y_{\boldsymbol{z}}$ as in equation (5), one can also compute the discrepancy between $f_\theta(\boldsymbol{z})$ and the "soft" probability $f_\theta(\boldsymbol{x}_i)$ in the absence of ground-truth label on augmented samples as in (Xie et al., 2019). In the following, We use superscript "hard" for the loss in equation (5) as

$$R_\theta^{\text{hard}}(\mathbb{P}_\theta^*,\boldsymbol{x}_i) = \sum_{\boldsymbol{z}\in B(\boldsymbol{x}_i)}\mathbb{P}_\theta^*(\boldsymbol{z}\mid\boldsymbol{x}_i)\ell(f_\theta(\boldsymbol{z}),y_{\boldsymbol{z}})) - \lambda_P\mathbb{P}_\theta^*(\boldsymbol{z}\mid\boldsymbol{x}_i)\log\left(|B(\boldsymbol{x}_i)|\mathbb{P}_\theta^*(\boldsymbol{z}\mid\boldsymbol{x}_i)\right), \quad (7)$$

to distinguish with the following objective which uses the "soft probability":

$$\begin{aligned}R_\theta^{\text{soft}}(\mathbb{P}_\theta^*,\boldsymbol{x}_i) = \ell(f_\theta(\boldsymbol{x}_i),y_{\boldsymbol{x}_i}) + \lambda_T\sum_{\boldsymbol{z}\in B(\boldsymbol{x}_i);\boldsymbol{z}\neq\boldsymbol{x}_i}\big(\mathbb{P}_\theta^*(\boldsymbol{z}\mid\boldsymbol{x}_i)\ell(f_\theta(\boldsymbol{z}),f_\theta(\boldsymbol{x}_i))\\ - \lambda_P\mathbb{P}_\theta^*(\boldsymbol{z}\mid\boldsymbol{x}_i)\log\left(|B(\boldsymbol{x}_i)|-1\right)\mathbb{P}_\theta^*(\boldsymbol{z}\mid\boldsymbol{x}_i)\big).\end{aligned} \quad (8)$$

The two terms in $R_\theta^{\text{soft}}(\mathbb{P}_\theta^*,\boldsymbol{x}_i)$ respectively correspond to the loss on original training samples $\boldsymbol{x}_i$ and the reweighted loss on the augmented samples. The reweighted loss promotes a small discrepancy between the augmented samples and the original training sample. $\lambda_T > 0$ is the coefficient used to balance the two loss terms, and $\mathbb{P}_\theta^*(\boldsymbol{z}\mid\boldsymbol{x}_i)$ is defined similar to (6) as

$$\mathbb{P}_\theta^*(\boldsymbol{z}\mid\boldsymbol{x}_i) = \frac{\exp\left(\frac{1}{\lambda_P}\ell(f_\theta(\boldsymbol{z}),f_\theta(\boldsymbol{x}_i))\right)}{\sum_{\boldsymbol{z}\in B(\boldsymbol{x}_i);\boldsymbol{z}\neq\boldsymbol{x}_i}\exp\left(\frac{1}{\lambda_P}\ell(f_\theta(\boldsymbol{z}),f_\theta(\boldsymbol{x}_i))\right)}. \quad (9)$$

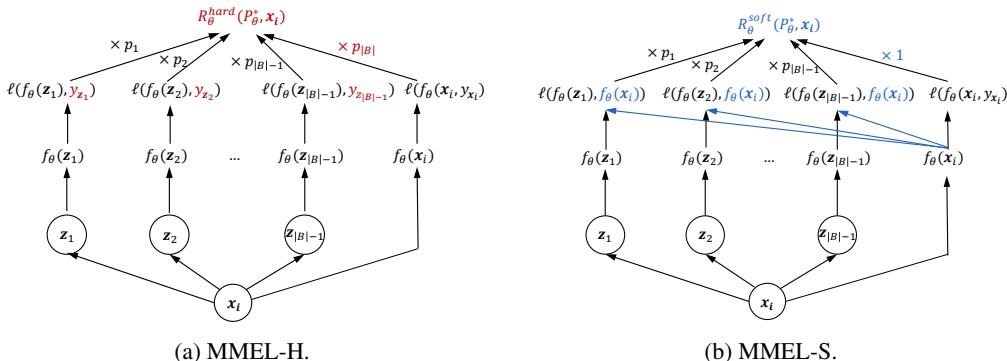

(a) MMEL-H.  (b) MMEL-S.

Figure 1: MMEL with two types of losses. Figure (1a) is the hard loss (7) with probability computed using (6) while Figure (1b) is the soft loss (8) with the probabilities computed using (9).

---

**Algorithm 1** Minimize the Maximal Expected Loss (MMEL)

---

**Input:** Training set $\{(\boldsymbol{x}_1, y_{\boldsymbol{x}_1}), \cdots, (\boldsymbol{x}_N, y_{\boldsymbol{x}_N})\}$, batch size $S$, learning rate $\eta$, number of training iterations $T$, $R_\theta$ equals $R_\theta^{\mathrm{hard}}$ or $R_\theta^{\mathrm{soft}}$.

1: **for** $i$ in $\{1, 2, \cdots, N\}$ **do**                          ▷ *generate augmented samples*
2:     Generating $B(\boldsymbol{x}_i)$ using some data augmentation method.
3: **end for**
4: **for** $t = 1, \cdots, T$ **do**                          ▷ *minimize the maximal expected loss*
5:     Randomly sample a mini-batch $\mathcal{S} = \{(\boldsymbol{x}_{i_1}, y_{\boldsymbol{x}_{i_1}}), \cdots, (\boldsymbol{x}_{i_S}, y_{\boldsymbol{x}_{i_S}})\}$ from training set.
6:     Fetch the augmented samples $B(\boldsymbol{x}_{i_1}), B(\boldsymbol{x}_{i_2}), \cdots, B(\boldsymbol{x}_{i_S})$.
7:     Compute $\mathbb{P}_\theta^*$ according to (6) or (9).
8:     Update model parameters $\theta_{t+1} = \theta_t - \frac{\eta}{S} \sum_{\boldsymbol{x} \in \mathcal{S}} \nabla_\theta R_\theta(\mathbb{P}_\theta^*, \boldsymbol{x})$.
9: **end for**

---

The two losses are shown in Figure 1. Summing over all the training samples, we get the two kinds of reweighted training objectives.

**Remark 4.** *The proposed MMEL-S tries to reduce the discrepancy between $f_\theta(\boldsymbol{z})$ and $f_\theta(\boldsymbol{x}_i)$ for $\boldsymbol{z} \in B(\boldsymbol{x}_i)$. However, if the prediction $f_\theta(\boldsymbol{x}_i)$ is inaccurate, such misleading supervision for $\boldsymbol{z}$ may lead to the degraded performance of MMEL-S. More details are in Appendix B.*

### 3.3 EXAMPLE: MMEL IMPLEMENTATION ON NATURAL LANGUAGE UNDERSTANDING TASKS

In this section, we elaborate on implementing the proposed method using textual data in natural language understanding tasks as an example. Our method is separated into two phases. In the first phase, we generate augmented samples. Then in the second phase, with these augmented samples, we update the model parameters under these augmented samples with respect to the hard reweighted loss (7) or the soft counterpart (8). The generation and training procedure can be decoupled, and the augmented samples are offline generated in the first phase by only once. On the other hand, in the second phase, since we have the explicit solution of weights on augmented samples and the multiple forward and backward passes on these augmented samples can be computed in parallel, the whole training time is similar to the regular training counterpart for an appropriate number of augmented samples. The whole training process is shown in Algorithm 1.

**Generation of Textual Augmented Data.** Various methods have been proposed to generate augmented samples for textual data. Recently, large-scale pre-trained language models like BERT (Devlin et al., 2019) and GPT-2 (Radford et al., 2019) learn contextualized representations and have been used widely in generating high-quality augmented sentences (Jiao et al., 2019; Kumar et al., 2020). In this paper, we use a pre-trained BERT trained from masked language modeling to generate augmented samples. For each original input sentence, we randomly mask $k$ tokens. Then we do a forward propagation of the BERT to predict the tokens in those masked positions by greedy search. Details can be found in Algorithm 2 in Appendix C.

**Mismatching Label.** For $R_\theta^{\text{hard}}$ in equation (7), the loss term $\ell(f_\theta(z), y_z)$ on augmented sample $z \in B(x_i)$ for some $x_i$ relies on its label $y_z$. Unlike image data, where conventional augmentation methods like random crop and horizontal flip of an image do not change its label, substituting even one word in a sentence can drastically change its meaning. For instance, suppose the original sentence is *"She is my daughter"*, and the word "She" is masked. The top 5 words predicted by the pre-trained BERT are "This, She, That, It, He". Apparently, for the task of linguistic acceptability task, replacing "She" with "He" can change the label from linguistically "acceptable" to "non-acceptable". Thus for textual input, for the term $\ell(f_\theta(z), y_z)$ in hard loss (7), instead of directly setting $y_z$ as $y_{x_i}$ (Zhu et al., 2020), we replace $y_z$ with the output probability of a trained teacher model. On the other hand, for the soft loss in equation (8), if an augmented sample $z \in B(x_i)$ is predicted to a different class from $x_i$ by the teacher model, it is unreasonable to still minimize the discrepancy between $f_\theta(z)$ and $f_\theta(x_i)$. In this case, we replace $f_\theta(x_i)$ in the loss term $\lambda_T \sum_{z \in B(x_i); z \neq x_i} \mathbb{P}_\theta^*(z \mid x_i) \ell(f_\theta(z), f_\theta(x_i))$ with the output probability from the teacher model.

## 4 EXPERIMENTS

In this section, we evaluate the efficacy of the proposed MMEL algorithm with both hard loss (MMEL-H) and soft loss (MMEL-S). Experiments are conducted on both the image classification tasks `CIFAR-10` and `CIFAR-100` (Krizhevsky et al., 2014) with the ResNet Model (He et al., 2016), and the General Language Understanding Evaluation (GLUE) tasks (Wang et al., 2019) with the BERT model (Devlin et al., 2019).

### 4.1 EXPERIMENTS ON IMAGE CLASSIFICATION TASKS.

**Data.** `CIFAR` (Krizhevsky et al., 2014) is a benchmark dataset for image classification. We use both `CIFAR-10` and `CIFAR-100` in our experiments, which are colorful images with 50000 training samples and 10000 validation samples, but from 10 and 100 object classes, respectively.

**Setup.** The model we used is ResNet (He et al., 2016) with different depths. We use random crop and horizontal flip (Krizhevsky et al., 2012) to augment the original training images. Since these operations do not change the augmented sample label, we directly adopt the original training sample label for all its augmented samples. Following (He et al., 2016), we use the SGD with momentum optimizer to train each model for 200 epochs. The learning rate starts from 0.1 and decays by a factor of 0.2 at epochs 60, 120 and 160. The batch size is 128, and weight decay is 5e-4. For each $x_i$, $|B(x_i)| = 10$. The $\lambda_P$ of the KL regularization coefficient is 1.0 for both MMEL-H and MMEL-S. The $\lambda_T$ in equation (8) for MMEL-S is selected from {0.5, 1.0, 2.0}.

We compare our proposed MMEL with conventional training with data augmentation (abbreviated as "Baseline(DA)") under the same number of epochs. Though MMEL can be computed efficiently in parallel, the proposed MMEL encounters $|B(x_i)| = 10$ times more training data. For fair comparison, we also compare with two other baselines that also use 10 times more data: (i) naive training with data augmentation but with 10 times more training epochs compared with MMEL (abbreviated as "Baseline(DA+Long)"). In this case, the learning rate accordingly decays at epochs 600, 1200 and 1600; (ii) training with data augmentation under the framework of MMEL but with uniform weights on the augmented samples (abbreviated as "Baseline(DA+UNI)").

**Main Results.** The results are shown in Table 1. As can be seen, for both `CIFAR-10` and `CIFAR-100`, MMEL-H and MMEL-S significantly outperform the Baseline(DA), with over 0.5 points higher accuracy on all four architectures. Compared to Baseline(DA+Long), the proposed MMEL-H and MMEL-S also have comparable or better performance, while being much more efficient in training. This is because our backward pass only computes the gradient of the weighted loss instead of the separate loss of each example. Compared to Baseline(DA+UNI) which has the same computational cost as MMEL-H and MMEL-S, the proposed methods also have better performance. This indicates the efficacy of the proposed maximal expected loss based reweighting strategy.

We further evaluate the proposed method on larege-scale dataset `ImageNet`(Deng et al., 2009). The detailed results are in Appendix B.

Table 1: Performance of ResNet on `CIFAR-10` and `CIFAR-100`. The time is the training time measured on a single NVIDIA V100 GPU. The results of five independent runs with "mean ($\pm$std)" are reported, expected for "Baseline(DA + Long)" which is slow in training.

| dataset | Model | Baseline(DA) | | Baseline(DA+Long) | | Baseline(DA+UNI) | | MMEL-H | | MMEL-S | |
|---|---|---|---|---|---|---|---|---|---|---|---|
| | | acc | time | acc | time | acc | time | acc | time | acc | time |
| CIFAR-10 | ResNet20 | 92.53($\pm$0.10) | 0.7h | **93.27** | 6.7h | 93.00($\pm$0.16) | 2.9h | 93.16($\pm$0.03) | 2.9h | 93.10($\pm$0.18) | 2.9h |
| | ResNet32 | 93.46($\pm$0.21) | 0.7h | **94.43** | 7.2h | 94.11($\pm$0.33) | 4.3h | 94.31($\pm$0.07) | 4.3h | 93.93($\pm$0.05) | 4.3h |
| | ResNet44 | 93.92($\pm$0.10) | 0.8h | 94.11 | 8.3h | 94.30($\pm$0.18) | 5.7h | **94.70**($\pm$0.14) | 5.7h | 94.48($\pm$0.08) | 5.7h |
| | ResNet56 | 93.96($\pm$0.20) | 1.1h | 94.12 | 10.6h | 94.62($\pm$0.18) | 7.0h | **94.85**($\pm$0.15) | 7.0h | 94.64($\pm$0.03) | 7.0h |
| CIFAR-100 | ResNet20 | 68.95($\pm$0.56) | 0.7h | 69.45 | 6.7h | 68.89($\pm$0.06) | 2.9h | **70.01**($\pm$0.07) | 2.9h | 70.00($\pm$0.07) | 2.9h |
| | ResNet32 | 70.66($\pm$0.16) | 0.7h | 71.98 | 7.2h | 71.59($\pm$0.10) | 4.3h | **72.51**($\pm$0.07) | 4.3h | 72.57($\pm$0.20) | 4.3h |
| | ResNet44 | 71.43($\pm$0.30) | 0.8h | 72.83 | 8.3h | 72.30($\pm$0.38) | 5.7h | **73.18**($\pm$0.31) | 5.7h | 72.89($\pm$0.16) | 5.7h |
| | ResNet56 | 72.22($\pm$0.26) | 1.1h | 73.09 | 10.6h | 73.44($\pm$0.13) | 7.0h | **74.20**($\pm$0.24) | 7.0h | 73.89($\pm$0.15) | 7.0h |

**Varying the Number of Augmented Samples.** One hyperparameter of the proposed method is the number of augmented samples $|B(\boldsymbol{x}_i)|$. In Table 2, we evaluate the effect of $|B(\boldsymbol{x}_i)|$ on the CIFAR dataset. We vary $|B(\boldsymbol{x}_i)|$ in $\{2, 5, 10, 20\}$ for both MMEL-H and MMEL-S with other settings unchanged. As can be seen, the performance of MMEL improves with more augmented samples for small $|B(\boldsymbol{x}_i)|$. However, the performance gain begins to saturate when $|B(\boldsymbol{x}_i)|$ reaches 5 or 10 for some cases. Since a larger $|B(\boldsymbol{x}_i)|$ also brings more training cost, we should choose a proper number of augmented samples rather than continually increasing it.

Table 2: Performance of MMEL on `CIFAR-10` and `CIFAR-100` with ResNet with varying $|B_{\boldsymbol{x}_i}|$. Here "MMEL-*-$k$" means training with MMEL-* loss with $|B(\boldsymbol{x}_i)| = k$. The results are averaged over five independent runs with "mean($\pm$std)" reported.

| dataset | Model | Baseline(DA) | MMEL-*-2 | | MMEL-*-5 | | MMEL-*-10 | | MMEL-*-20 | |
|---|---|---|---|---|---|---|---|---|---|---|
| | | | H | S | H | S | H | S | H | S |
| CIFAR-10 | ResNet20 | 92.53($\pm$0.10) | 92.77($\pm$0.01) | 92.91($\pm$0.21) | 93.11($\pm$0.13) | 92.89($\pm$0.05) | 93.16($\pm$0.03) | 93.10($\pm$0.18) | **93.57**($\pm$0.04) | 93.18($\pm$0.08) |
| | ResNet32 | 93.46($\pm$0.21) | 93.85($\pm$0.16) | 93.88($\pm$0.18) | 94.20($\pm$0.18) | 93.88($\pm$0.14) | 94.31($\pm$0.07) | 93.93($\pm$0.05) | **94.39**($\pm$0.09) | 93.89($\pm$0.17) |
| | ResNet44 | 93.92($\pm$0.10) | 94.18($\pm$0.12) | 93.87($\pm$0.13) | 94.51($\pm$0.13) | 94.35($\pm$0.07) | **94.70**($\pm$0.14) | 94.48($\pm$0.08) | **94.70**($\pm$0.20) | 94.39($\pm$0.11) |
| | ResNet56 | 93.96($\pm$0.20) | 94.29($\pm$0.08) | 94.43($\pm$0.05) | 94.78($\pm$0.09) | 94.56($\pm$0.15) | 94.85($\pm$0.15) | 94.64($\pm$0.03) | **95.01**($\pm$0.12) | 94.62($\pm$0.12) |
| CIFAR-100 | ResNet20 | 68.95($\pm$0.56) | 69.46($\pm$0.24) | 70.00($\pm$0.36) | 69.73($\pm$0.21) | 69.88($\pm$0.18) | 70.01($\pm$0.07) | 70.00($\pm$0.07) | 69.89($\pm$0.09) | **70.05**($\pm$0.23) |
| | ResNet32 | 70.66($\pm$0.16) | 71.50($\pm$0.09) | 71.37($\pm$0.30) | 72.41($\pm$0.18) | 71.73($\pm$0.16) | 72.51($\pm$0.07) | **72.57**($\pm$0.20) | 72.25($\pm$0.12) | 72.00($\pm$0.12) |
| | ResNet44 | 71.43($\pm$0.30) | 72.58($\pm$0.08) | 72.42($\pm$0.24) | **73.38**($\pm$0.07) | 72.92($\pm$0.15) | 73.18($\pm$0.31) | 72.89($\pm$0.16) | 73.23($\pm$0.18) | 72.77($\pm$0.15) |
| | ResNet56 | 72.22($\pm$0.26) | 73.11($\pm$0.36) | 73.33($\pm$0.30) | 73.95($\pm$0.04) | 73.47($\pm$0.14) | **74.20**($\pm$0.24) | 73.89($\pm$0.15) | 74.10($\pm$0.05) | 73.53($\pm$0.12) |

## 4.2 Results on Natural Language Understanding Tasks

**Data.** GLUE is a benchmark containing various natural language understanding tasks, including textual entailment (`RTE` and `MNLI`), question answering (`QNLI`), similarity and paraphrase (`MRPC`, `QQP`, `STS-B`), sentiment analysis (`SST-2`) and linguistic acceptability (`CoLA`). Among them, `STS-B` is a regression task, `CoLA` and `SST-2` are single sentence classification tasks, while the rest are sentence-pair classification tasks. Following (Devlin et al., 2019), for the development set, we report Spearman correlation for `STS-B`, Matthews correlation for `CoLA` and accuracy for the other tasks. For the test set for `QQP` and `MRPC`, we report "F1".

**Setup.** The backbone model is BERT$_{\text{BASE}}$ (Devlin et al., 2019). We use the method in Section 3.3 to generate augmented samples. For the problem of mismatching label as described in Section 3.3, we use a BERT$_{\text{BASE}}$ model fine-tuned on the downstream task as teacher model to predict the label of each generated sample $\boldsymbol{z}$ in $B(\boldsymbol{x}_i)$. For each $\boldsymbol{x}_i$, $|B(\boldsymbol{x}_i)| = 5$. The fraction of masked tokens for each sentence is 0.4. The $\lambda_P$ of the KL regularization coefficient is 1.0 for both MMEL-H and MMEL-S. The $\lambda_T$ in equation (8) for MMEL-S is 1.0. The other detailed hyperparameters in training can be found in Appendix D.

The derivation of MMEL in Section 3 is based on the classification task, while `STS-B` is a regression task. Hence, we generalize our loss function accordingly for regression tasks as follows. For the hard loss in equation (7), we directly replace $y_{\boldsymbol{z}} \in \mathbb{R}$ with the prediction of teacher model on $\boldsymbol{z}$. For the soft loss (8), for each entry of $f_\theta(\boldsymbol{x}_i)$ in loss term $\lambda_T \sum_{\boldsymbol{z} \in B(\boldsymbol{x}_i); \boldsymbol{z} \neq \boldsymbol{x}_i} \mathbb{P}_\theta^*(\boldsymbol{z} \mid \boldsymbol{x}_i) \text{MSE}(f_\theta(\boldsymbol{z}), f_\theta(\boldsymbol{x}_i))$, we replace it with the prediction of teacher model if the difference between them is larger than 0.5.

Similar to Section 4.1, We compare with three baselines. However, we change the first baseline to naive training without data augmentation (abbreviated as "Baseline") since data augmentation is not used by default in NLP tasks. The other two baselines are similar to those in Section 4.1: (i) "Baseline(DA+Long)" which fine-tunes BERT with data augmentation with the same batch size; and

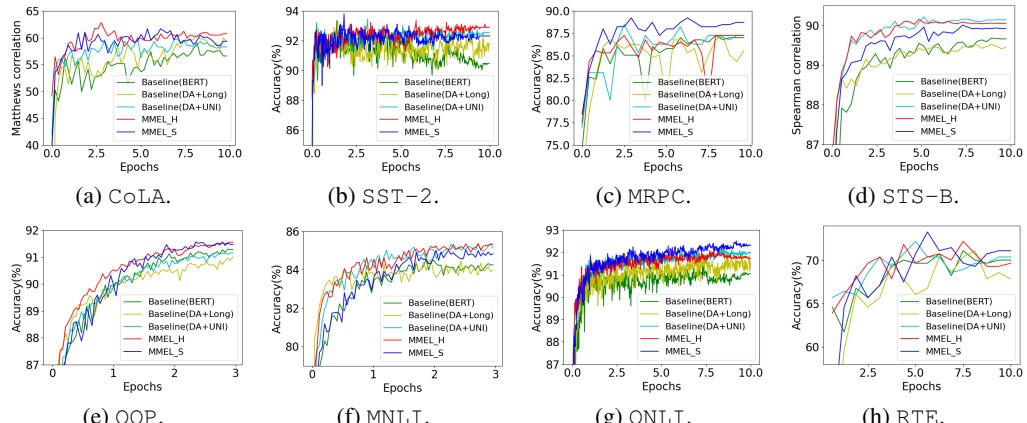

Figure 2: Development set results on BERT$_{\text{BASE}}$ model with different loss functions.

(ii)"Baseline(DA+UNI)" which fine-tunes BERT with augmented samples by using average loss. We also compare with another recent data augmentation technique SMART (Jiang et al., 2020b).

Table 3: Development and test sets results on the BERT$_{\text{BASE}}$ model. The training time is measured on a single NVIDIA V100 GPU. The results of Baseline, Baseline(DA+UNI), MMEL-H and MMEL-S are obtained by five independent runs with "mean($\pm$std)" reported.

| | Method | CoLA 8.5k | SST-2 67k | MRPC 3.7k | STS-B 7k | QQP 364k | MNLI-m/mm 393k | QNLI 108k | RTE 2.5k | Avg | Time |
|---|---|---|---|---|---|---|---|---|---|---|---|
| Dev | Baseline | 59.7($\pm$0.61) | 93.1($\pm$0.38) | 87.0($\pm$0.56) | 89.7($\pm$0.34) | 91.1($\pm$0.12) | 84.6($\pm$0.28)/85.0($\pm$0.37) | 91.7($\pm$0.17) | 69.7($\pm$2.3) | 83.5($\pm$0.27) | 21.5h |
| | Baseline(DA+Long) | 61.5 | 93.3 | 88.0 | 89.8 | 91.1 | 84.8/85.3 | 92.0 | **73.3** | 84.3 | 107.5h |
| | Baseline(DA+UNI) | 61.1($\pm$0.75) | 93.1($\pm$0.17) | 87.9($\pm$0.63) | 90.0($\pm$0.14) | 91.1($\pm$0.03) | 84.8($\pm$0.36)/85.1($\pm$0.26) | 91.9($\pm$0.16) | 71.8($\pm$1.02) | 84.1($\pm$0.14) | 31.6h |
| | SMART | 59.1 | 93.0 | 87.7 | 90.0 | 91.5 | **85.6/86.0** | 91.7 | 71.2 | 84.0 | - |
| | MMEL-H | **62.1**($\pm$0.55) | 93.1($\pm$0.14) | 87.7($\pm$0.20) | **90.4**($\pm$0.14) | **91.5**($\pm$0.07) | 85.3($\pm$0.06)/85.5($\pm$0.06) | 92.2($\pm$0.10) | 72.3($\pm$0.85) | 84.5($\pm$0.11) | 31.8h |
| | MMEL-S | **62.1**($\pm$0.55) | **93.5**($\pm$0.23) | **88.4**($\pm$0.73) | **90.4**($\pm$0.14) | **91.5**($\pm$0.04) | 85.2($\pm$0.05)/85.6($\pm$0.02) | **92.4**($\pm$0.12) | 71.9($\pm$0.24) | **84.6**($\pm$0.11) | 32.4h |
| Test | Baseline | 51.6($\pm$0.73) | 93.3($\pm$0.21) | 88.0($\pm$0.55) | 85.8($\pm$0.88) | 71.3($\pm$0.32) | 84.6($\pm$0.19)/83.8($\pm$0.30) | 91.1($\pm$0.35) | 67.4($\pm$1.30) | 79.6($\pm$0.09) | 21.5h |
| | Baseline(DA+Long) | 52.0 | 93.3 | **88.8** | **86.7** | 71.3 | 84.4/83.9 | 90.9 | 69.6 | 80.1 | 107.5h |
| | Baseline(DA+UNI) | 52.4($\pm$1.50) | 92.3($\pm$0.52) | 87.7($\pm$0.72) | 85.8($\pm$0.70) | 71.5($\pm$0.44) | 84.6($\pm$0.31)/83.6($\pm$0.46) | 90.6($\pm$0.43) | 68.8($\pm$1.76) | 79.7($\pm$0.50) | 31.6h |
| | MMEL-H | **53.6**($\pm$0.90) | 93.4($\pm$0.05) | 88.3($\pm$0.21) | 86.6($\pm$0.45) | **72.4**($\pm$0.05) | 84.9($\pm$0.19)/**84.5**($\pm$0.15) | **91.5**($\pm$0.11) | 69.8($\pm$0.52) | **80.5**($\pm$0.17) | 31.8h |
| | MMEL-S | 52.5($\pm$0.43) | **93.5**($\pm$0.16) | 88.3($\pm$0.15) | 86.1($\pm$0.07) | 72.1($\pm$0.10) | **85.0**($\pm$0.23)/84.2($\pm$0.15) | 91.4($\pm$0.31) | **69.9**($\pm$0.54) | 80.3($\pm$0.10) | 32.4h |

**Main Results.** The development and test set results on the GLUE benchmark are shown in Table 3. The development set results for the BERT baseline are from our re-implementation, which is comparable or better than the reported results in the original paper (Devlin et al., 2019). The results for SMART are taken from (Jiang et al., 2020b), and there are no test set results in (Jiang et al., 2020b). As can be seen, data augmentation significantly improves the generalization of GLUE tasks. Compared to the baseline without data augmentation (Baseline), MMEL-H or MMEL-S consistently achieves better performance, especially on small datasets like CoLA and RTE. Similar to the observation in the image classification task in Section 4.1, the proposed MMEL-H and MMEL-S are more efficient and have better performance than Baseline(DA+Long). MMEL-H and MMEL-S also outperform Baseline(DA+UNI), indicating the superiority of using the proposed reweighting strategy. In addition, our proposed method also beats SMART in both accuracy and efficiency because they use PGD-$k$ (Madry et al., 2018) to construct adversarial augmented samples which requires nearly $k$ times more training cost. Figure 2 shows the development set accuracy across over the training procedure. As can be seen, training with MMEL-H or MMEL-S converges faster and has better accuracy except SST-2 and RTE where the performance is similar.

**Effect of Predicted Labels.** For the augmented samples from same origin, we use a fine-tuned task-specific BERT$_{\text{BASE}}$ teacher model to predict their labels as mentioned in Section 3.3 to handle the problem of mismatching label. In Table 4, we show the comparison between using the label of the original sample and using predicted labels. As can be seen, using the predicted label significantly improves the performance. By comparing with the results in Table 3, using the label of the original sample even hurts the performance.

Table 4: Effect of using the predicted label. Development set results are reported.

| Method | Label | CoLA | SST-2 | MRPC | STS-B | QQP | MNLI-m/mm | QNLI | RTE | Avg |
|--------|-------|------|-------|------|-------|-----|-----------|------|-----|-----|
| MMEL-H | Original | 48.8 | 91.5 | 79.2 | 80.3 | 88.6 | 80.4/79.8 | 88.8 | 65.3 | 78.1 |
| MMEL-H | Predicted | 62.8 | 93.3 | 87.5 | 90.4 | 91.6 | 85.4/85.6 | 92.1 | 72.2 | 84.5 |
| MMEL-S | Original | 56.6 | 91.7 | 85.8 | 81.6 | 90.0 | 81.9/81.3 | 89.9 | 61.0 | 80.0 |
| MMEL-S | Predicted | 61.7 | 93.8 | 89.2 | 90.2 | 91.6 | 85.0/85.5 | 92.5 | 73.3 | 84.7 |

## 5 CONCLUSION

In this work, we propose to minimize a reweighted loss over the augmented samples which directly considers their implicit impacts on the loss. Since we can not access the optimal reweighting strategy, we propose to minimize the supremum of the loss under all reweighting strategies, and give a closed-form solution of the optimal weights. Our method can be applied on top of any data augmentation methods. Experiments on both image classification tasks and natural language understanding tasks show that the proposed method improves the generalization performance of the model, while being efficient in training.

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

# A PROOF OF THEOREM 1

*Proof.* For any given $\boldsymbol{x}_i$ and $B(\boldsymbol{x}_i)$, we aim to find $\mathbb{P}_\theta(\cdot \mid \boldsymbol{x}_i)$ on $B(\boldsymbol{x}_i)$ such that

$$\max_{\mathbb{P}_\theta(\cdot|\boldsymbol{x}_i)} \sum_{\boldsymbol{z} \in B(\boldsymbol{x}_i)} \mathbb{P}_\theta(\boldsymbol{z} \mid \boldsymbol{x}_i)\ell(f_\theta(\boldsymbol{z}), y_{\boldsymbol{z}})) - \lambda_P \mathbb{P}_\theta(\boldsymbol{z} \mid \boldsymbol{x}_i) \log\left(|B(\boldsymbol{x}_i)|\mathbb{P}_\theta(\boldsymbol{z} \mid \boldsymbol{x}_i)\right)$$
$$\text{s.t.} \sum_{\boldsymbol{z} \in B(\boldsymbol{x}_i)} \mathbb{P}_\theta(\boldsymbol{z} \mid \boldsymbol{x}_i) = 1. \tag{10}$$

Since the objective is convex, by Lagrange multiplier method, let

$$\mathcal{L}(\mathbb{P}_\theta, \lambda) = \sum_{\boldsymbol{z} \in B(\boldsymbol{x}_i)} \mathbb{P}_\theta(\boldsymbol{z} \mid \boldsymbol{x}_i)\ell(f_\theta(\boldsymbol{z}), y_{\boldsymbol{z}})) - \lambda_P \mathbb{P}_\theta(\boldsymbol{z} \mid \boldsymbol{x}_i) \log\left(|B(\boldsymbol{x}_i)|\mathbb{P}_\theta(\boldsymbol{z} \mid \boldsymbol{x}_i)\right)$$
$$+ \lambda \left( \sum_{\boldsymbol{z} \in B(\boldsymbol{x}_i)} \mathbb{P}_\theta(\boldsymbol{z} \mid \boldsymbol{x}_i) - 1 \right). \tag{11}$$

From $\nabla_{\mathbb{P}_\theta}\mathcal{L}(\mathbb{P}_\theta, \lambda) = \nabla_\lambda\mathcal{L}(\mathbb{P}_\theta, \lambda) = 0$, for any pairs of $\boldsymbol{z}_u, \boldsymbol{z}_v \in B(\boldsymbol{x}_i)$, we have

$$\ell(f_\theta(\boldsymbol{z}_u), y_{\boldsymbol{z}_u}) - \lambda_P(\log|B(\boldsymbol{x}_i)| + 1 + \log\mathbb{P}_\theta(\boldsymbol{z}_u \mid \boldsymbol{x}_i))$$
$$= \ell(f_\theta(\boldsymbol{z}_v), y_{\boldsymbol{z}_v}) - \lambda_P(\log|B(\boldsymbol{x}_i)| + 1 + \log\mathbb{P}_\theta(\boldsymbol{z}_v \mid \boldsymbol{x}_i)). \tag{12}$$

Hence we have

$$\mathbb{P}_\theta(\boldsymbol{z}_v \mid \boldsymbol{x}_i) = \mathbb{P}_\theta(\boldsymbol{z}_u \mid \boldsymbol{x}_i) \exp\left( \frac{\ell(f_\theta(\boldsymbol{z}_v), y_{\boldsymbol{z}_v}) - \ell(f_\theta(\boldsymbol{z}_u), y_{\boldsymbol{z}_u})}{\lambda_P} \right). \tag{13}$$

Summing over $\boldsymbol{z}_v \in B(\boldsymbol{x}_i)$, we have

$$\mathbb{P}_\theta(\boldsymbol{z}_u \mid \boldsymbol{x}_i) \sum_{\boldsymbol{z}_v \in B(\boldsymbol{x}_i)} \exp\left( \frac{\ell(f_\theta(\boldsymbol{z}_v), y_{\boldsymbol{z}_u}) - \ell(f_\theta(\boldsymbol{z}_u), y_{\boldsymbol{z}_u})}{\lambda_P} \right) = 1. \tag{14}$$

The proof completes. $\qquad\square$

# B MMEL ON LARGE-SCALE DATASET

In this section, we evaluate the proposed method MMEL on large-scale image classification task `ImageNet`(Deng et al., 2009).

**Data.** `ImageNet` is a benchmark dataset which contains colorful images with over 1 million training samples and 50000 validation samples from 1000 categories.

**Setup.** The model we used is ResNet for `ImageNet` with three different depths (He et al., 2016). All these experiments are conducted for 100 epochs, and the learning rate decays at epochs 30, 60, and 90. We set batch size as 256, and $|B(\boldsymbol{x}_i)| = 10$ for each $\boldsymbol{x}_i$. The other experimental settings follow Section 4.1, expect for the following hyperparameters. We compare the proposed method with "Baseline(DA)".

**Main Results.** The results are shown in Table 5. From the results, the proposed MMEL-H improves the performance of the model for all three depths. However, the proposed MMEL-S is beaten by the baseline method. We speculate this is due to the relatively larger proportion of inaccurate prediction of original training samples on the large-scale dataset. More specifically, as in equation (8), for each augmented sample $\boldsymbol{z} \in B(\boldsymbol{x}_i)$, the proposed MMEL-S encourages the model to fit the output of original training sample $f_\theta(\boldsymbol{x}_i)$. However, the accuracy of the original training samples in the `ImageNet` dataset can not reach 100% e.g., about 80% for ResNet50 on `ImageNet`. The inaccurate prediction $f_\theta(\boldsymbol{x}_i)$ can be a misleading supervision for augmented sample $\boldsymbol{z} \in B(\boldsymbol{x}_i)$, leading to degraded performance of the proposed MMEL-S. Thus, we suggest using the MMEL-H if the accuracy of the original training samples is relative low.

Table 5: Performance of ResNet on `ImageNet`.

| dataset | Model | Baseline(DA) | MMEL-H | MMEL-S |
|---------|-------|--------------|--------|--------|
| | ResNet18 | 69.76 | **70.48**(+0.72) | 68.52(-1.24) |
| `ImageNet` | ResNet34 | 73.30 | **74.38**(+1.08) | 72.33(-0.97) |
| | ResNet50 | 76.15 | **76.53**(+0.38) | 74.82(-1.33) |

## C  GENERATING AUGMENTED SAMPLES FOR TEXTUAL SAMPLES

In this section, we elaborate the procedure of generating augmented sentences using greedy-based and beam-based method for a sequence. For each original input sentence, we randomly mask $k$ tokens (which is obtained by rounding the product of masking ratio and length of the sequence to the nearest number) and then we do a forward propagation of the BERT to predict the tokens in those masked positions using greedy search. The detailed procedure is shown in Algorithm 2. We also use beam search (Yang et al., 2018) to generate augmented data. The details of beam search can be referred to (Yang et al., 2018). For sentence-pair tasks, we treat the two sentences separately and generate augmented samples for each of them.

---

**Algorithm 2** Augmented Sample Generation by Greedy Search

---

**Input:** Pre-trained language model `BertModel`, original sentence $x$, number of augmented samples $|B(x)| - 1$, number of masked tokens $k$.
**Output:** Augmented samples $B(x) = \{z_1, z_2, \cdots, z_{|B(x)|-1}\}$.

1: Randomly sample $k$ positions $\{p_1, \cdots, p_k\}$ and get $x_{mask}$.
2: **for** $i = 1, 2, \cdots |B(x)| - 1$ **do**      ▷ *Generate the i-th augmented sample*
3:    $z_i \leftarrow x_{mask}$.
4:    $z_i[p_1] \leftarrow$ the $i$th most likely word predicted by `BertModel`$(z_i[p_1]|z_i)$.
5:    **for** $j$ in $\{2, 3. \cdots, k\}$ **do**
6:       $z_i[p_j] \leftarrow$ the most likely word predicted by `BertModel`$(z_i[p_j]|z_i)$.
7:    **end for**
8: **end for**

---

In the following, we vary the factors that may affect the quality of the generated augmented samples. These factors include

1. The number of masked tokens, which equals the replacement proportion multiplied with the sentence length. This affects the diversity of augmented samples, i.e., replacing a large proportion of tokens makes the augmented sample less similar to the original one.

2. Treating the two sentences separately in sentence-pair tasks when generating augmented examples, or concatenate them as a single sentence;

3. Different generation methods like greedy search (Algorithm 2) and beam search.

The results are shown in Table 6. As can be seen, compared with Baseline without data augmentation, MMEL-H and MMEL-S under all hyperparameter configurations have higher accuracy, showing the efficacy of data augmentation and the proposed reweighting strategy. There is no significant difference in using greedy search or beam search to generate the augmented samples. In this natural understanding task, training with augmented samples generated with proper larger replacement proportion (i.e., larger diversity) has slightly better performance. For sentence-pair tasks, treating the two sentences separately and generate augmented samples for each of them has slightly better performance. In the experiments in Section 4.2, we use Greedy search, masking proportion 0.4, and generate augmented sentence for each sentence in sentence-pair tasks.

## D  HYPERPARAMETERS FOR THE EXPERIMENT ON THE GLUE BENCHMARK.

The optimizer we used is AdamW (Loshchilov & Hutter, 2018). The hyperparameters of BERT$_{\text{BASE}}$ model are listed in Table 7.

Table 6: Ablation study on generating augmented samples on the GLUE benchmark. Development set results are reported.

| | Method | Separate sentence-pair | Replacement proportion | CoLA | SST-2 | MRPC | STS-B | QQP | MNLI-m/mm | QNLI | RTE | Avg |
|---|---|---|---|---|---|---|---|---|---|---|---|---|
| | | Baseline | | 59.0 | 93.3 | 87.5 | 89.8 | 91.3 | 84.6/85.0 | 91.4 | 71.1 | 83.7 |
| MMEL-H | Greedy | True | 0.2 | 60.2 | 93.1 | 87.7 | 90.0 | 91.5 | 85.4/85.6 | 92.4 | 71.5 | 84.1 |
| | Greedy | True | 0.4 | **62.8** | 93.3 | 87.5 | **90.4** | **91.6** | 85.4/85.6 | 92.1 | **72.2** | **84.5** |
| | Greedy | False | 0.2 | 60.2 | **93.6** | 87.0 | 89.8 | 91.5 | 85.4/85.7 | **92.5** | 69.3 | 84.0 |
| | Greedy | False | 0.4 | 61.8 | 92.7 | **88.0** | 90.0 | 91.5 | 85.3/85.5 | 92.2 | **72.2** | 84.4 |
| | Beam | True | 0.2 | 60.0 | 93.2 | 87.0 | 90.0 | 91.4 | 85.5/85.5 | 92.4 | 71.1 | 84.0 |
| | Beam | True | 0.4 | 60.8 | 93.1 | **88.0** | 90.3 | 91.3 | 85.3/85.5 | 92.3 | 70.3 | **84.5** |
| | Beam | False | 0.2 | 60.0 | 93.5 | 86.7 | 89.8 | 91.5 | **85.6/85.7** | 92.3 | 70.4 | 83.9 |
| | Beam | False | 0.4 | 60.8 | 93.1 | **88.0** | 90.0 | 91.4 | 85.3/85.5 | 92.3 | 70.4 | 84.1 |
| MMEL-S | Greedy | True | 0.2 | 61.0 | 93.1 | 87.0 | 89.5 | **91.6** | 85.5/85.8 | 92.1 | 71.8 | 84.2 |
| | Greedy | True | 0.4 | 61.7 | **93.8** | **89.2** | **90.2** | **91.6** | 85.0/85.5 | **92.5** | 73.3 | **84.7** |
| | Greedy | False | 0.2 | 61.0 | 93.6 | 86.5 | 89.9 | **91.6** | **85.6/86.2** | **92.5** | 69.3 | 84.0 |
| | Greedy | False | 0.4 | 61.8 | 93.0 | 87.7 | 90.0 | **91.6** | 85.1/85.6 | **92.5** | 72.2 | 84.4 |
| | Beam | True | 0.2 | **62.0** | 92.9 | 86.7 | 89.9 | 91.5 | 85.4/85.9 | **92.5** | 71.8 | 84.2 |
| | Beam | True | 0.4 | 61.0 | 93.0 | 87.7 | **90.2** | 91.4 | 85.2/85.5 | 92.2 | **73.6** | 84.0 |
| | Beam | False | 0.2 | 62.0 | 93.3 | 86.7 | 89.7 | **91.6** | 85.3/85.9 | 92.4 | 70.0 | 84.1 |
| | Beam | False | 0.4 | 61.0 | 93.1 | 88.2 | 89.7 | 91.4 | 85.2/85.7 | 92.2 | 72.2 | 84.3 |

Table 7: Hyperparameters of the BERT$_{\text{BASE}}$ model.

| Hyperparam | MMEL-H | MMEL-S |
|---|---|---|
| Learning Rate | 3e-5 | 3e-5 |
| Batch Size | 32 | 32 |
| Weight Decay | 0 | 0 |
| Hidden Layer Dropout Rate | {0, 0.1} | {0, 0.1} |
| Attention Probability Dropout Rate | {0, 0.1} | {0, 0.1} |
| Max Epochs (MNLI, QQP) | 3 | 3 |
| Max Epochs (Others) | 10 | 10 |
| Learning Rate Decay | Linearly | Linearly |
| Warmup Ratio | 0 | 0 |
| $\lambda_T$ | 1.0 | 1.0 |
| $\lambda_P$ | 1.0 | 1.0 |
| Number of Candidates ($|B(\boldsymbol{x}_i)|$) | 5 | 5 |

