# OpenReview forum: "Reweighting Augmented Samples by Minimizing the Maximal Expected Loss"
_ICLR.cc/2021/Conference — ICLR 2021 Poster_

### Official Review · AnonReviewer4 · 2020-10-27
**REWEIGHTING AUGMENTED SAMPLES BY MINIMIZING THE MAXIMAL EXPECTED LOSS**

**Rating:** 6
**Confidence:** 4

**Review:**

##########################################################################

Summary:


The paper investigate an interesting problem, improving generalization performance by leveraging augmented data.

##########################################################################

Reasons for score:


Overall, my vote lies on borderline and my main concern is the novelty of the work.


##########################################################################
Pros:
1. The paper studies an important problem and propose a method that is a mixture of weighted augmentation and adversarial loss.
2. The paper is written well and easy to follow.
- Figure 1 is a nice representation to help the reader to understand the loss function better.
3. The paper investigates the performance of the proposed method on two domains: computer vision and natural language processing. The authors also provide ablation studies to evaluate the role of parameters.

##########################################################################

Cons:


1- I have two main concerns:
a) The novelty of the method is questionable for me as I have already seen a very close idea in the following papers:
[1] https://ieeexplore.ieee.org/abstract/document/8658998
[2] https://arxiv.org/pdf/1712.04621.pdf
[3]https://arxiv.org/abs/1907.12934
In [1], the min-max (adversarial) framework used and the distribution over the data provides weighting augmentation.
In [2], the effectiveness of data augmentation in object recognition has been discussed.
In [3], the min max entropy has been leveraged for weakly supervised (pixel-level) localization.
....
It is unfortunate that this papers were not even cited.
b) Generalization aspect.
The authors of the paper presents their main objective to provide generalization while they have not provided comprehensive experiments or discussion to address this concern.

2- Although the proposed method provides experiments on computer vision and natural language processing, I still suggest the authors to conduct more experiments considering more datasets from computer vision domain.

3-It would be great if the author can what would be the advantages/ weaknesses of their approach with the references. As the proposed approach has high similarity with the previous works, the minimum requirement is reporting more experiments and compare their method with exist methods. This extra study would present how their approach affect the performance.

##########################################################################

Questions during rebuttal period:


Please address and clarify the cons above


#########################################################################
After rebuttal:

Dear Authors, Thanks for providing more details. I believe more discussion and experiments are required to present the difference of you method. As you have mentioned, one difference is in considering summation rather than maximization, so it would be required to know what would be the advantages/ weaknesses of this difference. How does it make any impact on the performance? I would increase my score considering the closed-form solution as a nice contribution and requiring more experiments and analysis on the discussed references.

Thanks!

---

> ### Author Response · Authors · 2020-11-23
> **To AnonReviewer2 Part 2**
>
> Q2.	“Generalization aspect. The authors of the paper presents their main objective to provide generalization while they have not provided comprehensive experiments or discussion to address this concern.”
>
> A2.	First of all, our method is highly theoretically motivated (please refer to Section 3.1). Minimizing the proposed MMEL loss enables the model to perform well under any reweighting strategy. Secondly, in Section 4, we conducted various experiments on both computer vision tasks and natural language tasks and discussed the empirical results. These results show that our proposed method has better accuracy results than the compared methods.
>
> Q3.	“Although the proposed method provides experiments on computer vision and natural language processing, I still suggest the authors to conduct more experiments considering more datasets from computer vision domain.”
>
> A3.	Below we added more results with more augmentation methods, more architectures and more data sets on the computer vision domain.
>
> (1)	Since our method can be applied on top of any data augmentation technique. We further combine our method with one widely-used data augmentation method Cutout [2] on WideResNet [3]. The baseline results on WideResNet are taken from [2]. The other settings are the same as Section 4.1 of our submission. As can be seen, applying both MMEL-H and MMEL-S on top of Cutout have better performance than the original Cutout.
>
> |     Method              |     C10_Wide28_10    |     C10_Wide40_2    |     C100_Wide28_10    |     C100_Wide40_2    |
> |-------------------------|----------------------|---------------------|-----------------------|----------------------|
> |     Baseline(Cutout)    |     96.9             |     95.9            |     81.6              |     74.8             |
> |     MMEL-H              |     97.0(+0.1)       |     96.75(+0.85)    |     82.68(+1.02)      |     79.97(+5.15)     |
> |     MMEL-S              |     97.38(+0.48)     |     96.65(+0.75)    |     82.06(+0.40)      |     78.26(+3.46)     |
>
> (2)	We also apply the proposed method on the ImageNet data set on ResNet. The results of the baseline are taken from https://pytorch.org/docs/stable/torchvision/models.html.
> Due to time and computation limit, we only report the results of the proposed method with hard loss (MMEL-H). We use random crop and horizontal flip (Krizhevsky et al., 2012) to augment the original training images. We directly adopt the label of the original training sample for all its augmented samples. For each original example, we generate 3 more augmented samples from it. The other training hyperparameters are the same as [4]. As can be seen, by reweighting the augmented samples, MMEL-H has higher performance than the baseline with original data augmentation, especially on ResNet34.
>
> |     Method          |     ResNet18-Imagenet    |     ResNet34-Imagenet    |     ResNet50-Imagenet    |
> |---------------------|--------------------------|--------------------------|--------------------------|
> |     Baseline(DA)    |     69.76                |     73.30                |     76.15                |
> |     MMEL-H          |     70.48(+0.72)         |     74.38(+1.08)         |     76.53(+0.38)         |
>
>
> Reference:
>
> [1]	S. Behpour, K. Kitani, and B. Ziebart. "Ada: Adversarial data augmentation for object detection." 2019 IEEE Winter Conference on Applications of Computer Vision (WACV). IEEE, 2019.
>
> [2]	D., Terrance, and G. W. Taylor. "Improved regularization of convolutional neural networks with cutout." arXiv preprint arXiv:1708.04552 (2017).
>
> [3]	Z. Sergey, and N. Komodakis. "Wide residual networks.” arXiv preprint arXiv:1605.07146 (2016).
>
> [4]	https://github.com/pytorch/examples/tree/master/imagenet
>
> [5] A. Krizhevsky, and I. Sutskever, and G. E. Hinton “ImageNet Classification with Deep Convolutional Neural Networks” In Advances in neural information processing systems, pp. 1097–1105, 2012”

---

> ### Author Response · Authors · 2020-11-23
> **To AnonReviewer2 Part 1**
>
> Thanks for your review. We address the concerns as follows.
>
> Q1.	“The novelty of the method is questionable for me as I have already seen a very close idea in the following paper: [1] https://ieeexplore.ieee.org/abstract/document/8658998 In [1], the min-max (adversarial) framework used and the distribution over the data provides weighting augmentation. It is unfortunate that this paper was not even cited.”
>
> A1.	Thanks for pointing out this reference. We summarize the differences between this paper and [1] as follows.
>
> (1)	The first difference is the training objective. [1] studies the problem of object detection, and a perturbed bounding box $y$ is an augmented sample. As the reviewer suggested, if we rewrite the expected loss for input $x$ (equation (1) in [1]) as $\sum_{y}P(y|x)\sum_{y^{\prime}}f(y^{\prime}|x)l(y^{\prime},y)$, $\sum_{y^{\prime}}f(y^{\prime}|x)l(y^{\prime},y)$ can be viewed as the loss on augmented sample $y$ (bounding box) and $P(y\mid x)$ is the weights on the augmented samples i.e. annotation distribution.
>
> Our main learning objective (3) is different from (2) in [1] because their weights $P(y| x)$ are independent of the output of the model (to be learned) $f(\cdot)$ given $P(y’|x)$. However, we compute the weights of the augmented samples according to the loss on them which depends on $f(\cdot)$.
>
> Mathematically, for a given original sample $x$, if we use the same symbols as [1], our objective in equation (4) can be rewritten as $\max_{P(y|x)}\sum_{y}P(y|x)\sum_{y^{\prime}}f(y^{\prime}|x)[l(y^{\prime}, y)]$. This is different from the term $\sum_{y^{\prime}}f(y^{\prime}|x)\max_{P(y|x)} \sum_{y}P(y|x)l(y^{\prime}, y)$ in objective (2) in [1] because one can not interchange the order between summation and maximization.
>
> (2)	The second difference is the solver, which is a direct cause of the difference of the objective. The authors in [1] use linear programming to iteratively compute annotation distribution $P(y\mid x)$( Algorithm 1 in [1]), while we have a closed-form solution of the weights of the augmented samples in equation (6).
>
> (3)	The third difference is the application scenario. While for object detection, data augmentation can be done by directly generating perturbed labels (structured output of object detection), this method is not straightforward to be extended to other domains. On the other hand, our formulation can generally be applied in many domains.
>
> (4)	The method in [1] is more similar to adversarial training because (i) selecting bounding boxes that are maximally different from the ground truth mimics the process of generating the adversarial augmented samples and (ii) this generation process is included in each iteration of the learning process. However, for our proposed method, (i) the generation of augmented samples in our proposed method is decoupled with the learning process, and (ii) it can generally be applied on top of any data augmentation, regardless of whether the augmented samples are adversarially generated or not. The second paragraph in Section 2 of the paper clarifies the difference between the proposed method and adversarial training. We also added a discussion of [1] in the revised manuscript.

---

### Official Review · AnonReviewer1 · 2020-10-28
**Proposes an efficient example reweighting scheme that in some experiments can improve the performance but not always.**

**Rating:** 7
**Confidence:** 4

**Review:**

Summary:
This paper proposes a simple scheme for training with multiple augmentations of training data in one iteration and reweighting the instances by their relative loss. As authors note in their related works, the idea of reweighting examples based on their relative loss has been widely studied in a variety of machine learning problems. In contrast, this work proposes the reweighting only within augmentations of a single sample. They derive their particular reweighting scheme by proposing an alternative risk (Eq. 3). The new objective is a function of both model parameters and the distribution of augmentations. They propose to find the model parameters that minimize the alternative risk for the hardest distribution of augmentations that maximizes their alternative risk (Eq. 4). Then they consider the distribution of augmentations that are a function of model parameters and input and show that for fixed model parameters, the optimal distribution on a fixed finite set of augmentations is determined by the softmax on the loss of the model for each augmented input. In section 3.2, they propose two variations of their loss using the ground-truth label to evaluate the loss (hard loss) versus using the prediction of the model for the original raw input (soft loss). In section 3.3, they propose specific considerations for augmenting text data. They provide experiments on image and text data with ablations studies.

Pros:
- Proposed methods are particularly good on large models (resnet44, resnet56)
- Figure 2 is particularly interesting because the curves for proposed methods seem less noisy in multiple datasets (CoLA, SST-2 mrpc (only soft), qnli).
- The method might show more advantage if tested in low data settings or small mini-batches. Have authors tried any setting with small training data? Or smaller mini-batches?

Cons:
- The method does not always beat DA+UNI that uses multiple augmentations but weights them uniformly. Although one could argue DA+UNI is also an interesting contribution as it can be more efficient than DA+Long for small mini-batches.
- Tables: no standard deviation is given? In particular, in Table 1, 0.5 can easily be within the standard deviation.
- Table 1: DA+UNI should have the same time as MMEL but for resnet56 it takes 7h while MMEL takes 4.5h to train
- It is not clear how the hyperparameter lambda is tuned? Was it done using cross-validation? In the appendix the lambda is given as 1. Is that the best value?
- Table 3: Results on CoLA are good but others not significant? Again, not clear based on missing standard deviation.

Additional Notes:
- Can you clarify the following sentence: “Remark 3: If we ignore the KL divergence term in equation (3), due to the equivalence of minimizing cross-entropy loss and MLE loss...”
- Is Eq 7. just Eq 5 but for a single example?
- Figure 1 is not really descriptive enough. Maybe make it one figure and use color and line style to show the difference.
- DA+UNI: does the implementation use bigger batch size to perform computations in parallel?

Typos:
Section 3.1: an uniform -> a uniform
Remark 2: regularizedr -> regularizer

==============
After rebutall:
I thank the authors and appreciate addressing all my concerns. I'm glad that the additional experiments with smaller mini-batches and smaller training set size provide more supporting evidence for the method. I encourage the authors to point to these results in the main body.

One more minor suggestion is regarding the sentence in Remark 3. The way I read the sentence the "equivalence of minimizing CE loss and MLE loss" is the reason we can remove the KL divergence term. But the authors' response seems to say it is for the 3rd part of the sentence. The authors' might want to rewrite that sentence to make it clear.

---

> ### Author Response · Authors · 2020-11-23
> **To AnonReviewer 1 Part 1**
>
> Thanks for your review. We address the concerns as follows.
>
> Q1. “The method might show more advantage if tested in low data settings or small mini-batches. Have authors tried any setting with small training data? Or smaller mini-batches?”
>
> A1. We added experiments with small training data and smaller mini-batches on natural language understanding task QNLI from the GLUE Benchmark.
>
> (1) For small training set, we randomly select a subset from the original training corpus of QNLI. The table below shows results with varying subset size of 10K, 20K, 40K, and 80K training samples. As can be seen, the proposed method has higher accuracies than the other baselines on all these small training sets. The performance gain over the Baseline is more obvious on smaller training sets.
>
> |     dataset size    |     Baseline    |     Baseline(DA+Long)    |     Baseline(DA+UNI)    |     MMEL-H    |     MMEL-S    |
> |---------------------|-----------------|--------------------------|-------------------------|---------------|---------------|
> |     10k             |     87.3        |     87.6                 |     89.3                |     89.8      |     89.5      |
> |     20k             |     88.1        |     88.5                 |     89.6                |     90.7      |     90.1      |
> |     40k             |     89.9        |     90.3                 |     91.1                |     91.5      |     91.5      |
> |     80k             |     91.0        |     91.4                 |     91.7                |     92.3      |     92.2      |
> |     108K            |     91.7        |     92.0                 |     91.9                |     92.2      |     92.4      |
>
> (2) For smaller batch size, besides the default batch size 32 used in the paper, we further conduct experiments on QNLI with smaller batch size 4, 8 and 16 in the table below. As can be seen, the proposed method is more efficient than Baseline(DA+Long) for all different batch size settings. The speedup gain is more obvious when the batch size is smaller than 16. In addition, decreasing batch size incurs more performance degradation on all three baselines than MMEL-H and MMEL-S.
>
> |     Batch-size    |     Baseline       |     Baseline(DA+Long)    |     Baseline(DA+UNI)    |     MMEL-H             |     MMEL-S        |
> |-------------------|--------------------|--------------------------|-------------------------|------------------------|-------------------|
> |                   |     acc    time    |     acc    time          |     acc    time         |     acc   time         |     acc   time    |
> |     4             |     90.5  5.67h    |     90.1   28.35h        |     91.0      5.78h     |     92.2 5.78h         |     92.0 5.78h    |
> |     8             |     91.3  2.83h    |     90.7   14.15h        |     91.4      2.97h     |     92.3 2.97h         |     92.2 2.97h    |
> |     16            |     91.2  2.40h    |     91.2   12.00h        |     91.7      2.52h     |     92.2 2.52h         |     92.1 2.52h    |
> |     32            |     91.6  2.19h    |     92.0   10.95h        |     91.9      3.43h     |     92.1 3.43h         |     92.2 3.43h    |
>
> Q2.	“The method does not always beat DA+UNI that uses multiple augmentations but weights them uniformly.”
>
> A2. (1)  As is also suggested by the reviewer, DA+UNI is also one of our contributions even though we set it as a baseline.
>
> (2) In addition, from Table 1, MMEL is much better than DA+UNI on image classification task, though the improvements in natural language understanding (NLU) tasks in Table 3 is less obvious. We speculate that this is because for textual data, augmented samples generated by word substitution can be much more diverse compared with augmented image data, and the number of augmented samples also greatly affects the accuracy as well as how to efficiently utilize these samples (i.e. sample re-weighting). Thus MMEL does not always beat DA+UNI on NLU tasks as they have the same number of augmented samples.
>
> Q3.	“Tables: no standard deviation is given? In particular…In particular, in Table 1, 0.5 can easily be within the standard deviation.”
>
> A3.	We added mean and std results from 5 repetitions on both CIFAR and GLUE benchmark in Tables 1 and 3. Please also refer to our reply to Q4 for AnonReviewer3.
>
> Q4.	“Table 1: DA+UNI should have the same time as MMEL but for resnet56 it takes 7h while MMEL takes 4.5h to train”
>
> A4.	Thanks for pointing this typo out. We fixed it in the revised version.

---

> > ### Author Response · Authors · 2020-11-23
> > **To AnonReviewer 1 Part 2**
> >
> > Q5.	“It is not clear how the hyperparameter lambda is tuned? Was it done using cross-validation? In the appendix the lambda is given as 1. Is that the best value?”
> >
> > A5.	In this paper, we simply set both $\lambda_{T}$ and $\lambda_{P}$ as 1 without any tuning. A more careful tuning on them may lead to better performance. For example, we tune $\lambda_{T}$ and $\lambda_{P}$ in [0.5, 1.0, 2.0, 5.0] on task CoLA from the GLUE benchmark.
> > Compared to the default setting ($\lambda_{T}=\lambda_{P}=1$), the best $\lambda_{T}$ and $\lambda_{P}$ respectively increase the accuracy from 61.7 to 62.8 and 63.5 for MMEL-S. The best $\lambda_{P}$ on MMEL-H (there is no $\lambda_{T}$ for MMEL-H) increases the accuracy from 62.1 to 62.8. Please note that the results of CoLA in Table 3 are averaged over five independent runs. For this particular run of this tuning task, the accuracy with default $\lambda_{T}=\lambda_{P}=1$ is 61.7 and 62.1 for MMEL-H and MMEL-S, respectively.
> >
> > Q6.	“Can you clarify the following sentence: “Remark 3: If we ignore the KL divergence term in equation (3), due to the equivalence of minimizing cross-entropy loss and MLE loss...”
> >
> > A6.	If we ignore the KL divergence term in equation (3), the objective becomes $\frac{1}{N}\sum_{i=1}^{N} E[\ell(f_{\theta}(z), y_{z})]$. It is the sum of negative expected log-likelihood if $\ell(\cdot, \cdot)$ is the cross-entropy loss (which is commonly used for classification tasks). Then, minimizing equation (3) is equivalent to maximizing a log-likelihood loss. Please see page 8 in [3] for a more rigorous derivation. Under this condition, the equivalence of MMEL and GEM are then explained in the latter part of Remark 3.
> >
> > Q7.	“Is Eq 7. just Eq 5 but for a single example?”
> >
> > A7.	Yes.
> >
> > Q8.	“Figure 1 is not really descriptive enough. Maybe make it one figure and use color and line style to show the difference.”
> >
> > A8.	Thanks for this suggestion. We find it hard to put the two type of losses in one figure, so we mark the differences between them with different colors as suggested in Figure 1 in the revised manuscript.
> >
> > Q9.	“DA+UNI: does the implementation use bigger batch size to perform computations in parallel?”
> >
> > A9.	Yes, similar to MMEL-H and MMEL-S described in Section 3.3.
> >
> > Q10.	 “Typos: Section 3.1: an uniform -> a uniform, Remark 2: regularizedr -> regularizer”
> >
> > A10.	 Thanks. We have corrected them in the revised manuscript.
> >
> > Reference:
> > [1] J. Martens. "New insights and perspectives on the natural gradient method." arXiv preprint arXiv:1412.1193 (2014).

---

### Official Review · AnonReviewer3 · 2020-10-28
**Reweighting review**

**Rating:** 7
**Confidence:** 3

**Review:**

##########################################################################

Summary:

The paper proposes a novel data augmentation method. In particular, it proposes a reweighted loss function that allows to find the optimal weighting of the augmented samples. The approach is tested on standard image and
language tasks and compared to mulitple alternative approaches.

##########################################################################

Reasons for score:


Overall, I vote for accepting. The approach is interesting, well motivated and clearly formalized. However the experiments raise some questions which I would like the authors to comment on in the rebuttal.


##########################################################################

Pros:

1. The paper has an interesting take on the important topic of data augmentation

2. It proposes a clever and theoretically well motivated way to incorporate augmented samples.

3. The base line models are chosen reasonable.

4. The paper is clearly structured and well written.

##########################################################################

Cons:

1. One major concern is the performance comparison of varying numbers of augmented samples. Table 2 shows that often a smaller amount of augmented samples perform better. However, if the reweighting is indeed optimal, adding augmented samples should not hurt the performance.


2. It is not clear when and why to use the hard or soft loss. In addition with the varying performance dependent on the number of augmentation samples this adds additional hyperparameters. The increase in performance might not be worth the increase in training time (see point 3).


3. Are the results significant?


##########################################################################

Questions during rebuttal period:


Please address and clarify the cons above


#########################################################################

---

> ### Author Response · Authors · 2020-11-23
> **To AnonReviewer3**
>
> Thanks for your review. We address the concerns as follows.
>
> Q1. “One major concern is the performance comparison of varying numbers of augmented samples. Table 2 shows that often a smaller amount of augmented samples perform better. However, if the reweighting is indeed optimal, adding augmented samples should not hurt the performance.”
>
> A1. The reviewer is correct that more augmented samples do not harm the performance. However, the performance saturates after a certain number of augmented samples, and adding more examples makes the performance fluctuate around this saturation point’s performance. This is also why often the best performance is not necessarily achieved at the largest number of augmented samples, but at a smaller number. We also discussed this phenomenon at the end of Section 4.1 in the submission. To make this observation more reliable, we added results with mean and std from five independent runs in Table 2 in the revised manuscript. As can be seen from this table, the improvements brought by increasing augmented samples saturate after a certain number of augmented samples.
>
> Q2. “It is not clear when and why to use the hard or soft loss.”
>
> A2.  (1) When: As is discussed in Section 3.2, the hard loss in equation (7) requires the hard label of each augmented sample, while the soft loss in equation (8) only requires the soft output probability from the model. Thus we use the soft loss when the augmented data ends up being far from the clean data and the appropriate label for it is unclear.
>
> (2) Why: In the absence of a reliable ground-truth label for the augmented data, using the hard loss which requires the ground-truth label may mislead the learning. As is also shown in Table 4, in natural language tasks of the GLUE benchmark, the augmented samples generated by word substitution usually have different semantic meanings (as explained in Section 3.3) from the original sample. In this case, if we do not use the teacher model to predict labels, using hard loss (MMEL-H Original) has much more severe loss degradation than using soft loss (MMEL-S Original).
>
> Q3. “In addition with the varying performance dependent on the number of augmentation samples this adds additional hyperparameters. The increase in performance might not be worth the increase in training time (see point 3).”
>
> A3.  From Table 2, the performance improves with more augmented samples up to some certain number (Please also refer to our reply to Q1). Thus ideally we only need to stop increasing the number of augmented samples when we meet this saturation point. Empirically, we find that using 5 augmented samples for natural language understanding tasks of the GLUE benchmark and 10 augmented samples for image classification tasks on CIFAR achieves a good balance between the accuracy and training time.
>
> Q4. “Are the results significant?”
>
> A4. We added mean and std results from 5 repetitions on CIFAR in Tables 1&2 and GLUE benchmark in Table 3 in the revised manuscript. The small std in most cases indicates the stability of the proposed method. These results also show that the proposed method consistently outperforms the Baseline (DA) and Baseline (DA+UNI), while being more efficient in training than Baseline(DA+Long).

---

### Author Response · Authors · 2020-11-23
**General response**

We thank all the reviewers for their insightful and valuable comments. Please refer to the response to each reviewer for detailed explanations. We have revised the manuscript as suggested by the reviewers. The main changes (highlighted in blue) we made include:
1.	In Section 4, we add mean and std results for both image classification task CIFAR and the natural language understanding tasks from the GLUE benchmark. We do not report results for DA+Long because its training takes too much time.
2.	In related work, we added some more discussion about the connection and difference between the proposed method and one reference mentioned by Reviewer2.

Thanks again for all the valuable comments and suggestions.

---

### Decision · Program_Chairs · 2021-01-07
**Final Decision**

**Decision:**

Accept (Poster)

**Comment:**

Dear authors,

The reviewers appreciated the insights provided by your paper and the strong results. Congratulations.
I encourage you to address the other points raised to make your final submission as complete as possible.